# Predicting Pancreatic Ductal Adenocarcinoma Occurrence Up to 10 Years in Advance Using Features of the Main Pancreatic Duct in Pre-Diagnostic CT Scans

**DOI:** 10.3390/cancers17111886

**Published:** 2025-06-04

**Authors:** Lixia Wang, Yu Shi, Touseef Ahmad Qureshi, Yibin Xie, Srinivas Gaddam, Linda Azab, Chaowei Wu, Yimeng He, Zengtian Deng, Sehrish Javed, Garima Diwan, Camila Lopes Vendrami, Alex Rodriguez, Katherine Specht, Christie Y. Jeon, Humaira Chaudhry, James L. Buxbaum, Joseph R. Pisegna, Vahid Yaghmai, Wolfram Goessling, Yasmin G. Hernandez-Barco, Frank H. Miller, Temel Tirkes, Stephen J. Pandol, Debiao Li

**Affiliations:** 1Biomedical Imaging Research Institute, Cedars-Sinai Medical Center, Los Angeles, CA 90048, USA; lixia.wang@cshs.org (L.W.); yu.shi@cshs.org (Y.S.); touseefahmad.qureshi@cshs.org (T.A.Q.); yibin.xie@cshs.org (Y.X.); linda.azab@cshs.org (L.A.); chaowei.wu@cshs.org (C.W.); yimeng.he@cshs.org (Y.H.); zengtian.deng@cshs.org (Z.D.); sehrish.javed@cshs.org (S.J.); garima.diwan@cshs.org (G.D.); 2Department of Bioengineering, University of California, Los Angeles, CA 90048, USA; 3Karsh Division of Gastroenterology and Hepatology, Department of Medicine, Cedars-Sinai Medical Center, Los Angeles, CA 90048, USA; srinivas.gaddam@cshs.org; 4Department of Radiology, Northwestern Memorial Hospital, Northwestern University Feinberg School of Medicine, Chicago, IL 60611, USA; camila.vendrami@northwestern.edu (C.L.V.); frank.miller@nm.org (F.H.M.); 5Division of Gastroenterology, Department of Medicine, Keck School of Medicine of the University of Southern California, Los Angeles, CA 90033, USA; alex.rodriguez@med.usc.edu (A.R.); james.buxbaum@med.usc.edu (J.L.B.); 6Division of Gastroenterology, Massachusetts General Hospital, Boston, MA 02114, USA; katcspecht@gmail.com (K.S.); wgoessling@mgh.harvard.edu (W.G.); yhernandez-barco@mgh.harvard.edu (Y.G.H.-B.); 7Department of Biomedical Science, Cedars-Sinai Medical Center, Los Angeles, CA 90048, USA; christie.jeon@cshs.org; 8Department of Radiology, Rutgers New Jersey Medical School, Newark, NJ 07103, USA; chaudhhu@njms.rutgers.edu; 9Division of Gastroenterology, Hepatology and Parenteral Nutrition, Department of Medicine VA Greater Los Angeles Healthcare System, Los Angeles, CA 90073, USA; jpisegna@mednet.ucla.edu; 10Department of Radiological Sciences, University of California, Irvine, CA 92868, USA; vyaghmai@hs.uci.edu; 11Department of Radiology and Imaging Sciences, Indiana University School of Medicine, Indianapolis, IN 46202, USA; atirkes@iupui.edu; 12Division of Gastroenterology and Hepatology, Department of Medicine, Cedars-Sinai Medical Center, Los Angeles, CA 90048, USA; stephen.pandol@cshs.org

**Keywords:** pancreatic ductal adenocarcinoma (PDAC), main pancreatic duct (MPD), radiomics, PDAC prediction, artificial intelligence, pre-diagnostic CT

## Abstract

Pancreatic ductal adenocarcinoma (PDAC) is a highly lethal cancer, and early detection is crucial for improving survival. This study investigates whether analyzing features from the main pancreatic duct (MPD) in abdominal CT scans can enhance the prediction of PDAC before diagnosis. By integrating MPD features such as diameter and volume with pancreatic radiomics, our model significantly improved prediction, allowing for the identification of high-risk individuals up to 10 years in advance. These findings suggest that incorporating MPD analysis into AI-driven imaging techniques may help predict and detect PDAC at an earlier, more treatable stage, potentially improving patient outcomes.

## 1. Introduction

Pancreatic ductal adenocarcinoma (PDAC) is the deadliest malignant tumor worldwide, ranking as the third leading cause of cancer death with a 5-year relative survival probability of only 13% [1]. The tumor is typically detected at an advanced stage, which limits therapeutic options, contributing to a poor prognosis. Early detection is critical as it has been shown to significantly improve overall survival rates [2]. Previous studies reported stage 1A PDACs are more frequently eligible for R0 resection and adjuvant chemotherapy, resulting in a 5-year survival rate exceeding 80% for these patients [3]. The prediction of PDAC in high-risk patients is gaining increasing attention. Age, abdominal pain, obesity, weight change, HbA1c, alanine transaminase change, new-onset diabetes mellitus (DM), and genomics are all significant for early-onset PDAC [4,5,6].

CT is commonly used for abdominal examination and PDAC detection. CT is also effective in identifying secondary main pancreatic duct (MPD) dilation, pancreatic disease, lymph node status and distant metastasis [7,8]. The majority of PDACs are believed to originate from the branches of pancreatic ducts near the MPD [9]. Even small PDACs have a high possibility of extension and invasion into the MPD. Recent molecular studies have shown that PDAC development occurs, at least in part, through the intraepithelial proliferation/dysplasia–cancer sequence. The repeated stimulation of the pancreatic ductal epithelium by various risk factors, such as chronic pancreatitis and precancerous diseases, can lead to irreversible malignant transformation [10]. The slight dilation of the MPD and changes in MPD morphology can serve as a predictor of early PDAC [11,12]. MPD dilation is often a secondary change resulting from tumors, calculi, and other disease states [13]. The radiological appearance of the MPD on CT or MRI in various diseases shows some overlap with PDAC from mass-forming lesions.

Radiomics based on CT images has shown high performance in tumor prediction [14,15], early detection, differentiation, staging, detecting occult liver metastasis, and predicting the outcome of PDAC [16,17,18,19,20,21]. Most imaging-based PDAC prediction studies have focused on diagnostic images. For example, Cao et al. developed a deep learning model based on diagnostic CTs and controls to detect and classify pancreatic lesions, with a sensitivity and specificity of 92.9% and 99.9%, respectively [16]. Studies that have examined pre-diagnostic images have evaluated them for intervals up to 3 years prior to PDAC diagnosis [4,12,15], which limits inference for more distant predictions beyond 3 years. Additionally, using CT-based radiomics derived from the MPD and pancreas to develop a model for PDAC prediction on pre-diagnostic CT images has not been investigated.

The widely employed imaging modality to visualize the pancreatic ductal anatomy is magnetic resonance cholangiopancreatography (MRCP), which is non-invasive and carries much fewer complication risks compared to endoscopic retrograde pancreatography (ERCP) and shows a strong positive correlation in the MPD diameter measured with a special pancreatic ultrasound [22]. But cost limits the use of MRCP in routine clinical surveillance applications.

In this paper, we hypothesize that adding MPD features including diameter and volume and radiomics based on pre-diagnostic CT images can improve PDAC prediction in high-risk patients as PDAC-related changes manifest both in the MPD and the surrounding parenchyma well before PDAC diagnosis. Additionally, the model combining features from both the MPD and the pancreas can improve the prediction of PDAC occurrence within distinct timeframes (6 months to 3 years, 3–6 years, and 6–10 years in advance) compared to models that analyze either the MPD or pancreas alone. This study aims to assess the utility of integrating diameter, volume, and radiomic features from the MPD with those from the pancreas on pre-diagnostic abdominal CT scans for predicting PDAC across different timeframes in advance.

## 2. Materials and Methods

### 2.1. Ethics Approval

This retrospective study, including three (control, pre-diagnostic, and diagnostic) cohorts, was approved by the Institutional Review Board (IRB) of each participating medical center. All subjects enrolled in this study were de-identified prior to the expert’s reading and segmentation.

### 2.2. Data Description

A total of 8 medical centers provided redacted images for this retrospective study. This multicenter retrospective study involved abdominal contrast enhanced CTs of three cohorts. Of the initial 500 evaluated subjects, 90 subjects were excluded due to image quality, CTs with non-contrast, arterial or delayed phases, focal lesions of the pancreas in the control cohort and pre-diagnostic cohort, and no follow-up results obtained either by imaging or pathology. For MPD segmentation, 89 cases with a non-visible MPD were further excluded, including 63 cases from the control cohort (117 with a visible MPD out of 170 cases), 26 cases from the pre-diagnostic cohort (104 with a visible MPD out of 130 cases), and 10 cases from the diagnostic cohort (100 with a visible MPD out of 110 cases).

Diagnostic set: This dataset included abdominal contrast-enhanced CT scans with the portal venous phase and radiologist’s reports indicating a tumor or mass within the pancreas, confirming PDAC by histopathology, prior to surgery or chemotherapy between January 2012 and December 2022. Patients who had a history of previous pancreatic surgery (e.g., Whipple or pancreatectomy procedures) were excluded. This cohort consisted of 100 abdominal CTs (50 females and 50 males and a median age of 69 years with an age range of 46–96 years) with histopathologically confirmed PDAC.

Pre-diagnostic abdominal contrast-enhanced CT were defined as scans conducted between 6 months and 10 years prior to PDAC diagnosis. For patients with multiple pre-diagnostic CT scans, the closest scan at least 6 months prior to the PDAC diagnostic date was selected. Pre-diagnostic CT was negative for PDAC, pancreatic cystic lesions such as mucinous cystic neoplasm (MCN), serous cystic neoplasm (SCN), and other kinds of benign and malignant tumors during radiological interpretation. Additionally, CT images with secondary imaging signs of PDAC, such as attenuation difference, biliary or pancreatic duct dilatation, or obstruction due to a tumor, were excluded.

This dataset comprised 104 abdominal CTs from subjects (52 females and 52 males and a median age of 66 years, with an age range of 41–93 years) who developed PDAC at least 6 months after the pre-diagnostic CT. The median interval between pre-diagnostic CTs and PDAC diagnosis was 3.02 years, ranging from 6 months to 10 years.

Control set: A control group matched to the cases with pre-diagnostic images by sex and age at the time of diagnosis (less than 3 years) was created from patients with abdominal CTs with a normal pancreas. These include abdominal contrast-enhanced CT scans taken for abdominal pain and other indications. The control dataset included individuals with no history of pancreatic surgery, diabetes mellitus (DM), MCN, SCN, other benign tumors or tumor-like lesions, or acute and chronic pancreatitis at the time of the CT scans. No focal or diffuse atrophy was detected by radiologists. This cohort consisted of abdominal CTs from 117 subjects (58 females and 59 males, and a median age of 63 years, with an age range of 40–90 years) without pancreatic disorders, confirmed by imaging and blood biomarkers.

Three hundred twenty-one subjects were finally enrolled in three cohorts. Sample size was determined based on the available data that met the inclusion criteria, and no formal power calculation was performed due to the retrospective nature of the study. The portal venous phase of abdominal contrast-enhanced CT was segmented in this study by one radiologist (LW). The portal venous phase was chosen because it provides a consistent enhancement of the pancreatic parenchyma, which improves the contrast between the pancreatic duct and surrounding tissue, thereby facilitating the accurate labeling of the MPD. It is also the most consistently acquired in routine abdominal imaging protocols, ensuring greater uniformity across the cohort and minimizing variability introduced by other phases [23]. The venous phase was obtained 60–75 s after the intravenous administration of a contrast agent of a weight-based dose, generally ranging from 75 to 100 mL for patients. CTs were performed on various scanner systems including Siemens, GE, Toshiba, and Philips, with a matrix of 512 × 512 and slice thicknesses of 2.5–5 mm. The detailed parameters are listed in Appendix A.

### 2.3. Image Segmentation and Radiomic Feature Extraction and Feature Reduction

The CTs of all subjects were downloaded and de-identified by the anonymization of Digital Imaging and Communication in Medicine (DICOM). Metadata including scanner equipment specifications, CT slice thickness, the field of view, and the acquisition matrix were extracted from DICOM headers. As the datasets were multi-center contrast-enhanced CT images, standardized normalization and a harmonization pipeline were performed to reduce inter-center variability. All CT studies were first loaded into ITK-snap Software (version 3.6.0; RRID: SCR_002010) and saved in the Neuroimaging Informatics Technology Initiative (NIFTI) format (RRID: SCR_003141). Next, all images were resampled to a uniform voxel size (e.g., 1 × 1 × 1 mm^3^) to ensure spatial consistency and optionally denoised to enhance image quality. Intensity normalization is applied by clipping Hounsfield Unit (HU) values to a predefined range (e.g., −50 to 200 for soft tissue) to address scanner differences. Histogram matching is performed to align intensity distributions across scans. Following this, harmonization is conducted using ComBat (RRID: SCR_010974) harmonization, a widely used statistical approach to correct scanner- and center-related biases while preserving biological variations in radiomic features.

Once the images are standardized, the segmentation of the MPD and pancreas is performed.

The MPD was manually segmented slice by slice in the portal venous phase by a board-certified abdominal oncologic radiologist (LW) with 25 years of experience. For each subject in the three cohorts, the visible portion of the MPD on each slice was labeled. The pancreas was auto-segmented using an abdominal organ segmentation model followed by manual correction to ensure its quality. The surrounding vessels, stents of the common bile duct, and accessory spleen were removed, while calcifications within the pancreas and MPD were retained.

After segmentation, radiomic features are extracted using the open-source Python (RRID: SCR_024202) package PyRadiomics (version 3.8.5; Python Software Foundation, Wilmington, DE; RRID: SCR_026019), capturing first-order, texture, and shape characteristics. In cases where portions of the MPD were invisible on CT images, we conducted a piecewise feature analysis on the visible segments. The visible portions of the MPD were segmented into distinct regions based on their anatomical characteristics. Radiomic features were calculated separately for each segmented region. These features were then analyzed to capture localized variations within the MPD. For the invisible segments, these features were excluded from the analysis to ensure that the results remained robust and free from potential bias introduced by missing data. To minimize bias, the radiologist who performed ductal and pancreatic segmentation and radiomic feature extraction was blinded to cohort assignments.

The maximum, minimum, mean, and standard deviation (SD) of the ductal diameter and the segmented volume of the MPD for each subject were measured using the Python package in a 2D transverse position.

A total of 107 radiomics features were extracted from the MPD and the pancreas alone for each subject, including 14 shape features, 18 first-order features, and 75 texture features. Texture features included 24 gray-level cooccurrence matrices (GLCMs), 16 gray-level run-length matrices (GLRLMs), 16 gray-level size zone matrices (GLSZMs), 14 gray-level dependence matrices (GLDMs), and 5 neighborhood gray-tone difference matrix (NGTDM) features.

### 2.4. Statistical Analysis

To manage a large number of radiomic features, an unpaired two-tailed Student’s *t*-test was first used to identify significant differences between control and pre-diagnostic features (*p* > 0.05). Normality was checked using the Shapiro–Wilk test, and equal variances were tested with Levene’s test. All analyses were performed using SPSS software, version 24 (IBM Corp., Armonk, NY, USA; RRID: SCR_002865). Next, highly correlated radiomic features (the threshold was 0.8) were identified and removed to eliminate redundant information. This reduced the feature space dimension. An ANOVA was performed to select radiomic features based on the classifier, using a recursive feature selection approach. An SVM classifier was used and hyperparameters were determined using 5-fold cross-validation on the training data.

Model performance was evaluated using receiver operating characteristic (ROC) curve analysis based on the diameter and volume of the MPD, features of the MPD and pancreas alone, and their combinations. The area under the ROC curve (AUC), accuracy, sensitivity, and specificity were calculated at a cutoff value that maximized the Youden index [24]. The DeLong test was performed to compare the AUC values from different models in predicting PDAC within each timeframe. *p* values less than 0.05 indicated statistically significant differences. The trends of all selected features in the model were also observed. The radiomics workflow is shown in Figure 1.

A one-way ANOVA was performed to investigate the differences in the diameter and volume of the MPD among the 3 cohorts using IBM SPSS statistics, version 24 (IBM Corp., Armonk, NY, USA; RRID:SCR_002865). Turkey’s test was used for post hoc multiple comparisons among the three groups, respectively. The Pearson correlation coefficient was calculated to observe the correlation between the ductal diameter measured on CT and MRCP.

## 3. Results

### 3.1. Demographic Information

There are 221 participants with a total 321 CT scans in the study. The average age at the time of the CT image was 66.2 years, ranging 40–96 years. The sample size and the demographic distribution are presented in Table 1. The distribution of time intervals between pre-diagnostic CT and PDAC occurrence is presented in Appendix A.

### 3.2. Trend of Diameters and Volumes for Control, Pre-Diagnostic, and Diagnostic Data

The sample size, maximum ductal diameter, minimum ductal diameter, mean ductal diameter, standard deviation, and volume for each scan group are listed in Table 2. All parameters displayed an increasing trend (Figure 2 and Figure 3). The one-way ANOVA indicated significant differences in the diameter and volume of the MPD among the groups (*p* < 0.05). When comparing between two groups, all parameters, except for the Min diameter and volume between control and pre-diagnostic cohorts, showed statistically significant differences (*p* < 0.05). Appendix A illustrates the increased trend in diameter from the control to pre-diagnostic and diagnostic cohorts validated with MRCP and demonstrates the correlation between the Max ductal diameter on CT and MRCP (Pearson correlation coefficient: r = 0.894).

### 3.3. Performance of Radiomic Models for PDAC Prediction Across Timeframes

Table 3 summarizes the performance of models based on the diameter and volume of the MPD, the addition of radiomic features of the MPD to the diameter and volume, radiomics features from the pancreas, and the combination of both the MPD and pancreas in predicting PDAC across different timeframes: 6 months–3 years, 3–6 years, 6–10 years and 6 months–10 years in advance. All models showed their highest predictive performance within 6 months–3 years in advance, with AUC values of 0.81, 0.88, 0.83, and 0.96 for the four models, respectively. Model performance varied for more distant prediction intervals over time, with AUC values of 0.77, 0.81, 0.81, and 0.94 for images taken 3–6 years in advance and 0.69, 0.75, 0.75, 0.84 for images taken 6–10 years in advance. For the overall pre-diagnostic cohort with 6 months to 10 years in advance, the AUC values of the three models were 0.71, 0.72, 0.77, and 0.83. Additional evaluation metrics are listed in Table 3.

The model of the MPD including radiomic features, diameter and volume performed comparably to the pancreas model in predicting PDAC for the three timeframes. The addition of MPD features to those of the pancreas significantly improves the model’s performance compared to the radiomics model of the pancreas alone for all timeframes (*p* < 0.05). Figure 4 shows ROC curves of four models based on the MPD including radiomic features, diameter and volume, the pancreas, and a combination of the MPD and pancreas for each time interval. All extracted features, selected feature coefficients, definitions, and clinical relevance are shown in Appendix A. Table 4 and Appendix A show comparisons of different model performances calculated by the Delong test in each time interval. Appendix A demonstrate the trends of selected features across the control, pre-diagnostic, and diagnostic cohorts. Appendix A shows the feature importance in the combined model of different timeframes using the SHAP method.

## 4. Discussion

To the best of our knowledge, this is the first study to investigate the predictive performance of radiomic features, diameter and volume from the MPD both independently and integrated with the radiomics of the pancreas in predicting PDAC occurrence across distinct timeframes: 6 months–3 years, 3–6 years, and 6–10 years and 6 months–10 years in advance. The pre-diagnostic and diagnostic CT images demonstrated a progressive increase in the diameter and volume of the MPD. Radiomic models derived from the MPD including radiomic features, the diameter and volume, the pancreas, and a combination of both exhibited the highest predictive performance within the 6 months–3 year intervals, with AUC values of 0.81, 0.88, 0.83, and 0.96, respectively, with attenuated accuracy over longer intervals. The addition of the MPD including the diameter and volume and radiomics improved the PDAC prediction AUC from 0.83 to 0.96 for subjects 6 months to 3 years in advance, from 0.81 to 0.94 for those 3–6 years in advance, and 0.75 to 0.84 for those 6–10 years in advance of diagnosis, respectively. These results suggested that MPD features are important biomarkers for predicting PDAC, even up to 10 years in advance, and should be included for PDAC prediction.

These findings align with previous results, which demonstrated high sensitivity in detecting MPD changes including slight dilation or narrowing [12,25]. Positive findings were significantly associated with the classifying of small PDAC and a normal pancreas. Tanaka et al. reported that 65% of pre-cancer cases had an MPD diameter larger than 2 mm on ultrasonography, and the mean diameter of the MPD in the pre-cancer group increased over time [26]. Our results showed that 72.1% (75/104) of pre-diagnostic cases had an MPD diameter larger than 2 mm and increased from the pre-diagnostic cohort to the diagnostic cohort. The measurements and trends in the MPD diameter and volume utilizing different protocols or scanners remained consistent, suggesting that the differences in imaging protocols did not significantly affect this finding. To avoid bias in radiomic features, we performed image harmonization and normalization before segmentation and extracting radiomic features. Anatomical variability, patient age, and tumor stage can significantly affect MPD visibility and delineation on CT. Variations in ductal anatomy, such as branching patterns or naturally narrow ducts, may reduce contrast with surrounding tissue, complicating segmentation. The MPD diameter tends to increase with age, and age-related changes like parenchymal atrophy or ductal ectasia can either improve visibility or introduce boundary ambiguity. In early or pre-diagnostic tumor stages, MPD changes are often subtle and hard to detect, while advanced tumors typically cause obstruction and upstream dilation, enhancing MPD visualization.

MRCP served as a complementary modality to CT, particularly in confirming subtle ductal changes. In cases where both CT and MRCP were performed within a close time interval (typically within 30 days), the maximum MPD diameter was compared at matched anatomical locations. Agreement was assessed using both qualitative and quantitative measures. High correlation was observed between two modalities, with a Pearson correlation coefficient of r = 0.894, indicating strong consistency. Notable discrepancies were primarily related to ductal morphology; MRCP offered a superior visualization of subtle features such as irregularities or mild dilation, which may be less apparent on CT. Nonetheless, the overall trend of increasing ductal size across disease stages was consistently observed on both imaging modalities.

MPD dilatation is noted in various pancreatic conditions and is an easily missed sign of tumor onset [4,27,28,29]. Chronic pancreatitis (CP), characterized by long-term and recurrent inflammation and pancreatic fibrosis, can lead to the narrowing and irregularity of the MPD, predisposing individuals to PDAC development [30,31]. Precursor lesions such as pancreatic intraepithelial neoplasia and pancreatic intraductal papillary mucinous neoplasm (IPMN) can contribute to MPD dilatation by stimulating the secretion of pancreatic fluid and blocking or compressing the MPD [32]. In some instances, PDAC tumors originate from the branch pancreatic duct and spread along the ductal system, causing ductal dilation, and occur before the development of a visible mass. The dense fibrotic stroma of non-cancerous pancreatic parenchyma can cause mechanical compression and the torturing of the MPD. Additionally, tumor-induced ischemia, neoplastic angiogenesis, and PDAC-associated cystic changes within the mass further exacerbate MPD dilatation by disrupting normal tissue architecture and blood flow within the pancreas.

Previous studies have highlighted the importance of predicting PDAC. Identifying individuals at a high risk of developing PDAC can enable early detection and the ongoing monitoring of high-risk factors and further improve the outcome [33,34]. In recent years, various efforts have been made to stratify PDAC risk using pre-diagnostic abdominal CT scans. Qureshi examined the anatomical characteristics of the entire pancreas through radiomic analysis on pre-diagnostic abdominal CT scans to predict PDAC within 6–36 months and achieved 85.7% accuracy [35]. Javed further extended the anatomical analysis to pancreatic subregions for a more precise PDAC prediction, resulting in a 3.6% improvement in classification accuracy [15]. Two subsequent studies adopted a similar approach and reached comparable conclusions [36,37]. However, these studies only predict PDAC occurrence within 3 years. There is a possibility of missing cases that slowly develop PDAC after 3 years. In our dataset, nearly half of the pre-diagnostic cases developed PDAC after 3 years.

Predicting PDAC occurrence across distinct timeframes (6 months–3 years, 3–6 years, and 6–10 years in advance) is crucial as it addresses the significant variability in disease progression, particularly in individuals with slower-evolving precursor lesions like PanINs and IPMNs [38,39]. Long-standing diabetes (>3 years) is associated with increased risk for PDAC [40]. Pancreatitis diagnosed beyond 5 years continues to present an elevated adjusted relative hazard of 2.02 for PDAC [41]. Early-stage pancreatic changes, which may remain undetected or asymptomatic within a shorter timeframe, can progress to invasive cancer over several years. This extended prediction interval up to 10 years allows for the identification of at-risk individuals who might not show immediate signs but could benefit from ongoing monitoring and early intervention strategies. By capturing these later-occurring cases, clinicians can implement timely preventive measures, ultimately improving patient outcomes and reducing mortality associated with PDAC.

The innovative finding in our study is using radiomic features of the MPD plus diameter and volume, the pancreas, and their combination to predict PDAC across different timeframes up to 10 years. The model combining radiomic features, diameter and volume from the MPD shows superior performance within 6 months–3 years (AUCs: 0.88) and promising results within 3–6 years and 6–10 years (AUCs: 0.81 and 0.75), highlighting the importance of the MPD in PDAC prediction for a shorter interval, likely due to the direct involvement of the MPD in the early pathophysiological changes associated with pancreatic cancer. Our findings also highlighted that the addition of radiomic features from the MPD improved PDAC prediction beyond the conventional diameter and volume model within 6 months–3 years in advance (AUCs from 0.81 to 0.88). Among all radiomic features, gldm_DependenceNonUniformity, glrlm_RunEntropy, glszm_SizeZoneNonUniformityNormalized, and glszm_SmallAreaEmphasis were selected as the most predictive radiomic features. These features measure the variability in dependence groups in the image, capture the randomness of run lengths in the image, calculate the variability in zone sizes, and emphasize fine-textured regions by highlighting the proportion of small homogeneous zones that reflect textural heterogeneity, microstructural complexity, and subtle tissue variations that may precede overt anatomical changes. Unlike diameter and volume, which primarily reflect macroscopic ductal dilation, these radiomic features provide a more granular assessment of tissue irregularities, offering an earlier and more precise detection of pre-diagnostic abnormalities that are likely influenced by factors such as changes in ductal contents and pressure, ductal dilatation, and scarring associated with PDAC, pancreatitis, and other pathological processes. The increasing trend observed in radiomic features from the control to pre-diagnostic and diagnostic cohorts further emphasizes the growing complexity and heterogeneity within the MPD area. The superior predictive performance of adding radiomic features from the MPD underscores the potential of radiomics in detecting pre-diagnostic abnormality from the MPD that are imperceptible in routine imaging. Gldm_DependenceNonUniformityNormalized, gldm_GrayLevelNonUniformity and glszm_GrayLevelNonUniformityNormalized from the pancreas are the most significant radiomic features for predicting PDAC within 6 months to 3 years in advance. DependenceNonUniformityNormalized reflects the variability in local intensity dependencies, suggesting increased tissue complexity, while GrayLevelNonUniformity and GrayLevelNonUniformityNormalized measure gray-level distribution irregularities, which may indicate early desmoplastic changes and altered pancreatic tissue organization. These subtle variations in tissue heterogeneity may relate to changes in microstructure like fibrosis and fatty infiltration [42] and correspond to early microstructural alterations associated with tumor initiation.

For predicting PDAC 3–6 years in advance, our study identified key radiomic features of both the main pancreatic duct (MPD) and the pancreas and significantly enhanced PDAC prediction when combining MPD and pancreas. From the MPD, the following features were selected including Max ductal diameter, firstorder_RobustMeanAbsoluteDeviation, which measures the mean distance of all intensity values from the mean value based on a subset of images with gray levels between the 10th and 90th percentile, glcm_SumEntropy, which represents the sum of intensity differences between neighboring pixels, glcm_SumSquares, which measures the variance in neighboring intensity-level pairs around the mean intensity level in the GLCM, gldm_GrayLevelVariance, which evaluates the variance in gray level intensities within the zones, glszm_GrayLevelNonUniformityNormalized, which measures the variability in gray-level intensity values in the image, normalized for size, glszm_SizeZoneNonUniformity, which assesses the variability of size zone volumes in the image, where a lower value suggests greater homogeneity in zone sizes, and ngtdm_Complexity, which measures the level of complexity in an image by evaluating the variations in intensity between a voxel and its neighboring voxels.

These features also reflect variations in textural heterogeneity, structural irregularities, and early microarchitectural changes in ductal tissue. Similarly, InterquartileRange, SumSquares, DependenceEntropy, and Sphericity from the pancreas, which measure tissue intensity variations, gray-level dispersion, and morphometric differences, are the most predictive features and are associated with early neoplastic alterations.

In predicting PDAC within 6–10 years in advance, the selected radiomic features from the MPD include firstorder_Skewness and glszm_SizeZoneNonUniformityNormalized, which capture the asymmetry in the intensity distribution within the MPD and quantify the variability in homogeneous intensity regions, indicating irregular tissue organization within the duct. Glcm_JointEnergy, gldm_DependenceNonUniformityNormalized, gldm_DependenceVariance, gldm_LargeDependenceEmphasis and glrlm_RunPercentage from the pancreas capture tissue heterogeneity, disorganization and altered glandular structure, a hallmark of tumor development.

The addition of radiomics of the MPD, including diameter and volume, to the pancreas showed the highest predictive performance for PDAC, with AUCs of 0.96, 0.94, 0.84, and 0.83, respectively, for the different pre-diagnostic intervals. The superior performance of this combined model outperformed the radiomic model based on the pancreas alone (*p* < 0.05), which suggests that integrating features from both anatomical structures that can capture complementary information is proven to be a promising approach in radiomics research and reinforces the hypothesis that PDAC-related changes manifest both in the MPD and the surrounding parenchyma well before PDAC diagnosis. The selected features reflect the early changes in the shape and tissue physiology environment, including the MPD and pancreas, to provide a more comprehensive evaluation and improve the robustness of predictive models. Our results showed consistency with the previous consensus that a combination of multi-region radiomics analysis can lead to higher predictive performance compared to single-region analysis. The model’s performance varied over longer time durations, with AUC values decreasing as the interval extended to 6–10 years in advance, which aligns with the biological and pathological progression of PDAC [43]. This trend potentially reflects the increasing heterogeneity and complexity of the disease over time, where early changes in the pancreatic tissue that are more easily captured by radiomic features may become less distinct as the disease progresses or as the precursor lesions evolve at different rates [44]. This approach not only improves early detection but also extends the predictive capacity into longer intervals, supporting the notion that different radiomic features may dominate at various stages of disease progression.

There are some limitations in our study. Firstly, this retrospective study benefits from having a larger sample size than previous efforts, but its power remains limited in detecting subtle feature differences that could further enhance discriminatory performance. To mitigate this, we utilized multicenter data, applied advanced statistical methods, and enhanced feature engineering to improve model robustness and reduce single-center bias. However, a key limitation is the absence of prospective external validation, which is essential to confirm the model’s generalizability in real-world settings. Future work will focus on acquiring prospective datasets to validate the model’s performance over time and ensure its generalization. Secondly, the MPD is a very detailed structure within the pancreas, and some parts of the MPD in CT images may be invisible, especially for control subjects, and result in disconnection when MPD segmentation is performed. We use piecewise feature analysis to extract meaningful radiomic characteristics from the visible portions of the MPD. Future work will focus on developing autosegmentation techniques of very detailed structures or other methods to extend the invisible portions of the MPD, improving segmentation consistency and feature extraction. Finally, the control cohort was confirmed to have no pancreatic disorders based on imaging and blood biomarkers, and genetic predispositions to pancreatic diseases were not available. Some control subjects may be at an elevated risk due to underlying genetic factors. Future studies should integrate genetic screening or family history analysis to further refine the control group and improve the robustness of comparisons.

## 5. Conclusions

Pancreatic radiomics of pre-diagnostic abdominal CT scans have been shown to predict PDAC occurrence within 3 years. In this work, we developed radiomic models to predict PDAC occurrence in various timeframes up to 10 years in advance. The addition of the radiomics of the MPD significantly improves the performance of PDAC prediction compared to the model using pancreas radiomics alone across different timeframes before diagnosis. A major challenge of including the MPD for PDAC prediction using CT scans is that the MPD is not always visible. Future work will explore autosegmentation techniques to improve reproducibility and extend the visibility of the MPD in cases where segmentation is challenging. Additionally, future research will validate these findings in larger and prospective cohorts and develop models incorporating age, gender, and ethnicity alongside the radiomics model, as well as other radiomic biomarkers associated with PDAC occurrence.

## Figures and Tables

**Figure 1 cancers-17-01886-f001:**
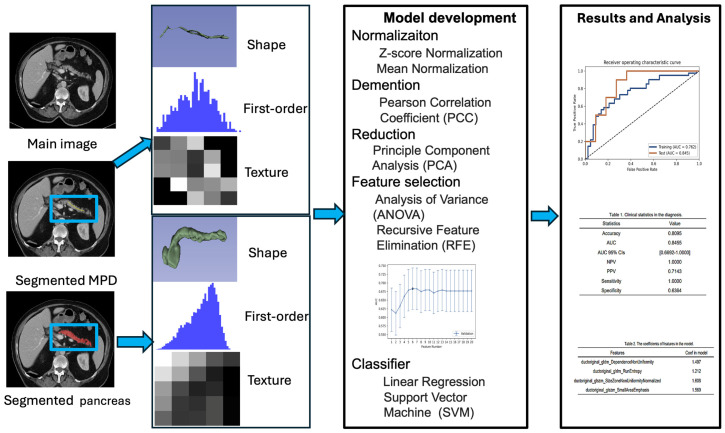
The workflow of radiomics analysis in this study.

**Figure 2 cancers-17-01886-f002:**
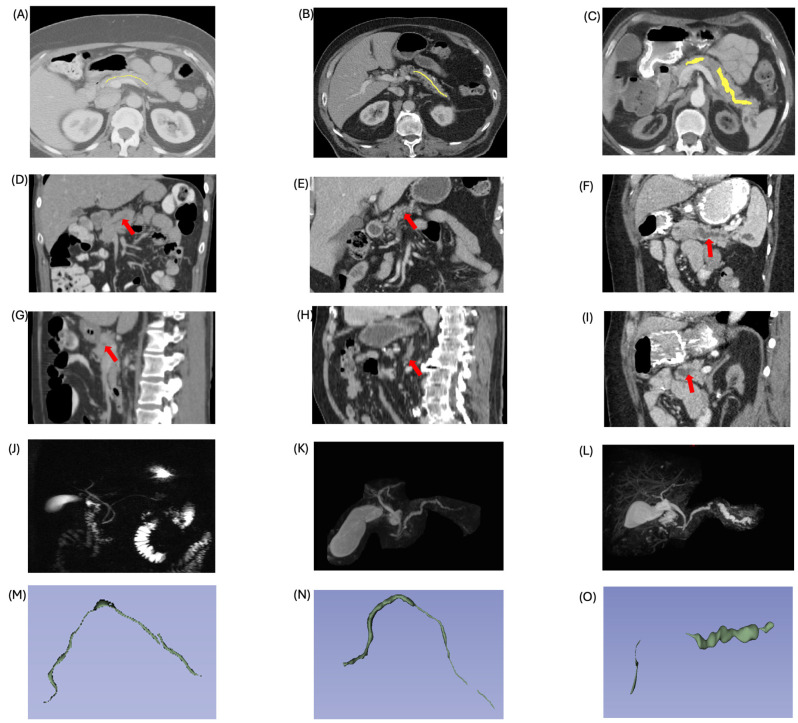
Main pancreatic duct (MPD) segmentation and imaging across disease progression. The top row (**A**–**C**) displays axial CT images with MPD segmentations in control (**A**), pre-diagnostic (**B**), and diagnostic (**C**) subjects. The yellow line within the pancreas represents the segmentation of the main pancreatic duct. From control to pre-diagnostic to diagnostic subjects, the diameter of main pancreatic duct gradually increase. The second row (**D**–**F**) presents corresponding coronal views, while the third row (**G**–**I**) shows sagittal views. The red arrows in the figures indicate the appearance of main pancreatic duct from different views. The fourth row (**J**–**L**) includes MRCP images of the same cases, and the fifth row (**M**–**O**) shows 3D reconstructions of the segmented MPD. The pre-diagnostic subject demonstrates slight MPD dilatation, evident in both MRCP and 3D views. In the diagnostic case with PDAC, segmentation and MRCP reveal a disrupted MPD, indicating tumor-induced blockage and upstream ductal dilation.

**Figure 3 cancers-17-01886-f003:**
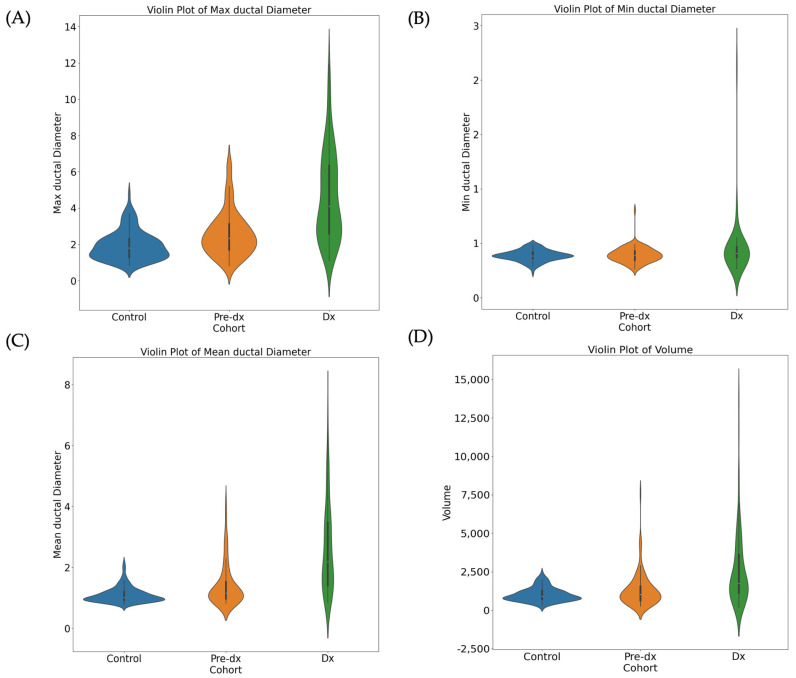
Violin plots of the Max ductal diameter (**A**), Min ductal diameter (**B**), mean ductal diameter (**C**) and volume (**D**) across the control (blue), pre-diagnostic (orange) and diagnostic cohorts (green). Each plot illustrates the distribution and central tendency (white dot = median) of ductal measurements within each cohort. The control cohort shows a wider distribution with a lowest median value in Max ductal diameter, Min ductal diameter, mean ductal diameter and volume, suggesting greater variability but overall smaller ductal sizes and volumes. The Pre-dx cohort shows a moderate spread and a median between the control cohort and Dx cohort, suggesting early structure changes in the MPD. The Dx cohort shows a narrower distribution and the highest median value, indicating more consistently elevated values with disease progression. Statistical analysis confirms significant differences and an increasing trend in ductal measurements from the control to pre-diagnostic and diagnostic cohorts. (Pre-dx: pre-diagnostic cohort; Dx: diagnostic cohort, MPD: main pancreatic duct).

**Figure 4 cancers-17-01886-f004:**
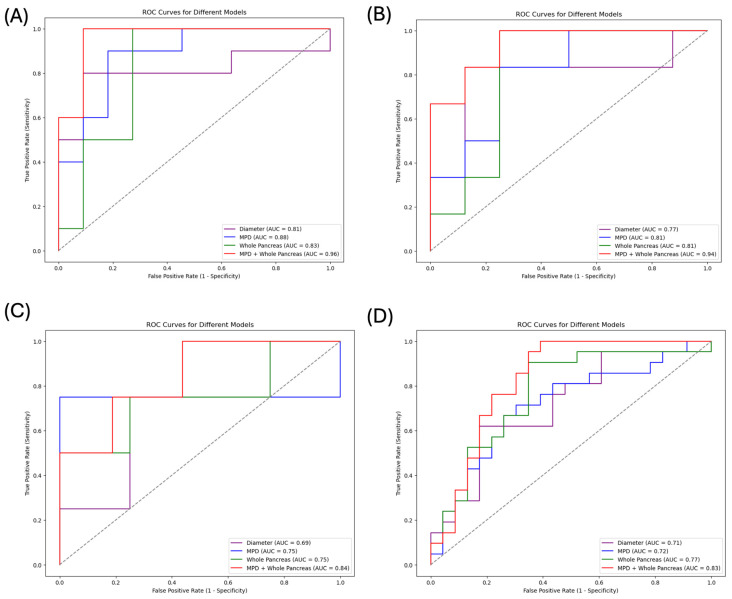
Performance analysis of radiomic models for predicting PDAC occurrence within 6 months–3 years (**A**), 3–6 years (**B**), 6–10 years (**C**), and 6 months–10 years (**D**). The graphs present receiver operating characteristic (ROC) curves for models based on the diameter and volume of main pancreatic duct (MPD), diameter and radiomic features of the MPD, the pancreas, and a combination of both. The purple line represents the model of the diameter and volume, the blue line corresponds to the model derived from the MPD, the green line corresponds to the model derived from the pancreas, and the red line represents the model based on the combination of the MPD and the pancreas. The diagonal dotted line represents the performance of a classifier that makes random guesses.

**Table 1 cancers-17-01886-t001:** Demographics of the participants included in this study.

Variable	Control	Pre-Diagnostic	Diagnostic
Subjects (CT scans)	117	104	100
Median age (range), in years	63 y (40–90 y)	66 y (41–93 y)	69 y (44–96 y)
Gender			
Female	58	52	50
Male	59	52	50
Duratio n between pre-diagnostic and diagnostic scan	Mean 3.02 years (0.60–10 years)		
6 months–3 years	51 cases (49.0%)		
3–6 years	32 cases (30.8%)		
6–10 years	21 cases (20.2%)		

**Table 2 cancers-17-01886-t002:** Diameters and volume of main pancreatic duct, and difference among three cohorts (0 = control cohort, 1 = pre-diagnostic cohort, 2 = diagnostic cohort).

Class	Sample Size	Max Ductal Diameter (mm)	Min Ductal Diameter (mm)	Mean Ductal Diameter (mm)	Standard Deviation	Ductal Volume (mm^3^)
Control	117	1.94 ± 0.80	0.89 ± 0.05	1.10 ± 0.25	0.31 ± 0.29	1017.41 ± 446.58
Pre-diagnostic	104	2.59 ± 1.24	0.89 ± 0.07	1.41 ± 0.70	0.48 ± 0.43	1296.21 ± 1061.06
Diagnostic	100	4.67 ± 2.49	0.97 ± 0.31	2.63 ± 1.46	1.07 ± 0.67	2583.22 ± 2335.84
*p*-value (ANOVA)		0.000	0.001	0.001	0.000	0.000
	0 vs. 1	0.009	0.961	0.032	0.023	0.334
	0 vs. 2	0.000	0.002	0.000	0.000	0.000
	1 vs. 2	0.000	0.006	0.000	0.000	0.000

**Table 3 cancers-17-01886-t003:** Diagnosis performance of models in classifying control and pre-diagnostic pancreas within distinct timeframes.

	Model	ACC	AUC	AUC (95% CIs)	SE	SP
6 months–3 years	Diameter + volume	0.86	0.81	0.59–1	0.80	0.91
	MPD	0.86	0.88	0.73–1	0.90	0.82
	Pancreas	0.86	0.83	0.65–1	1	0.73
	MPD + pancreas	0.95	0.96	0.89–1	1	0.91
3–6 years	Diameter + volume	0.86	0.77	0.46–1	0.83	0.88
	MPD	0.79	0.81	0.58–1	0.83	0.75
	Pancreas	0.86	0.81	0.56–1	0.83	0.75
	MPD + pancreas	0.86	0.94	0.82–1	1	0.75
6–10 years	Diameter + volume	0.75	0.69	0.25–1	0.75	0.75
	MPD	0.87	0.75	0.26–1	0.75	1
	Pancreas	0.75	0.75	0.35–1	0.75	0.75
	MPD + pancreas	0.88	0.84	0.53–1	0.75	1
6 months–10 years	Diameter + volume	0.73	0.71	0.56–0.87	0.62	0.83
	MPD	0.70	0.72	0.56–0.87	0.71	0.70
	Pancreas	0.77	0.77	0.62–0.91	0.90	0.65
	MPD + pancreas	0.80	0.83	0.70–0.96	0.86	0.70

(MPD = diameter + volume + radiomics of main pancreatic duct, ACC = accuracy, CI = confidence interval, SE = sensitivity, SP = specificity).

**Table 4 cancers-17-01886-t004:** The differences in AUCs between two models within each timeframe.

	Delong Test	*p* Value
6 months–3 years	Diameter vs. MPD	0.02
	MPD vs. pancreas	0.38
	MPD vs. MPD + pancreas	0.010
	Pancreas vs. MPD + pancreas	0.000
3–6 years	Diameter vs. MPD	0.08
	MPD vs. pancreas	0.34
	MPD vs. MPD + pancreas	0.002
	Pancreas vs. MPD + pancreas	0.01
6–10 years	Diameter vs. MPD	0.45
	MPD vs. pancreas	0.65
	MPD vs. MPD + pancreas	0.05
	Pancreas vs. MPD + pancreas	0.05
6 months–10 years	Diameter vs. MPD	0.76
	MPD vs. pancreas	0.000
	MPD vs. MPD+ pancreas	0.000
	Pancreas vs. MPD + pancreas	0.003

## Data Availability

The data presented in this study are available on request from the corresponding authors upon reasonable request.

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
