# Peer review of "Predicting Pancreatic Ductal Adenocarcinoma Occurrence Up to 10 Years in Advance Using Features of the Main Pancreatic Duct in Pre-Diagnostic CT Scans"

_cancers, 2025, doi:10.3390/cancers17111886_

Round 1

Reviewer 1 Report

Comments and Suggestions for Authors

This manuscript presents the results from CT imaging scans from 3 patient groups to assess if specific features of the Main Pancreatic Duct (MPD) can contribute to the early detection of the highly lethal Pancreatic ductal adenocarcinoma (PDAC) and improve patient survival.

Assessing these CT scans using learning algorithms and incorporating AI may have the potential to more accurately detect the early stages of PDAC non-invasively. The manuscript is interesting, and it will make valuable contribution in the field of cancer imaging. The manuscript is well written. However, some minor issues need to be corrected/clarified before publication.

  1. Although statistically significant due to the large number of samples, the mean differences in diameter between the control (1.1+0.25mm) pre diagnostic (1.41+0.7mm) and diagnostic (2.63+1.46) groups are small (less than 2mm). Is there any data that clearly measures the changes in the MPD diameter size over time?
  2. The graphs should be labelled correctly according to cohort measured (not class) in figure 3, A2, A4, A5, A6 and A7. The font in Figures 3, 4 and the supplementary section are blurry too small to read.
  3. Table 3 does not provide a clear comparison between control and pre-diagnostic pancreas as stated and needs to be adjusted to reflect the figure legend or made clearer.
  4. The values (%, number?) and time scales (days, months) and not specified throughout this manuscript and need be added.

Author Response

Thank you very much for taking the time to review this manuscript. Please find the detailed responses below and the corrections were highlighted in the resubmitted files. 

Reviewer 1

This manuscript presents the results from CT imaging scans from 3 patient groups to assess if specific features of the Main Pancreatic Duct (MPD) can contribute to the early detection of the highly lethal Pancreatic ductal adenocarcinoma (PDAC) and improve patient survival.

Assessing these CT scans using learning algorithms and incorporating AI may have the potential to more accurately detect the early stages of PDAC non-invasively. The manuscript is interesting, and it will make valuable contribution in the field of cancer imaging. The manuscript is well written. However, some minor issues need to be corrected/clarified before publication.

1. Although statistically significant due to the large number of samples, the mean differences in diameter between the control (1.1+0.25mm) pre diagnostic (1.41+0.7mm) and diagnostic (2.63+1.46) groups are small (less than 2mm). Is there any data that clearly measures the changes in the MPD diameter size over time?

Thank you for your comment. We acknowledge that the mean differences in MPD diameter between the control (1.10 ±â€¯0.25 mm), pre-diagnostic (1.41 ±â€¯0.70 mm), and diagnostic (2.63 ±â€¯1.46 mm) groups are numerically small.  So we analyzed Max ductal diameter and  Ductal volume as well and presented in Table2.  These parameters were observed a substantial increase across three groups, with a clear upward trend from control to pre-diagnostic to diagnostic cohort.

2. The graphs should be labelled correctly according to cohort measured (not class) in figure 3, A2, A4, A5, A6 and A7. The font in Figures 3, 4 and the supplementary section are blurry too small to read.

I appreciated your comments. We have revised these figures and updated the related labels to clearly reflect the respective cohorts measured (e.g., control, pre-diagnostic, diagnostic) and improved font size and resolution. These changes ensure the figure more accurately and improves clarity for the reader. The revised figures have been uploaded as per journal requirements. We appreciate your feedback, which helped us enhance the quality of our figures.

3. Table 3 does not provide a clear comparison between control and pre-diagnostic pancreas as stated and needs to be adjusted to reflect the figure legend or made clearer.

Thank you for your comments. Table 3 and all figure legend have been further clarified.

4. The values (%, number?) and time scales (days, months) and not specified throughout this manuscript and need be added.

Thank you for your careful review. We appreciated the observation regarding missing units and unclear time scales in several sections of the manuscript. We added appropriate units (e.g., percentages, absolute counts) to all reported values where needed. Time scales (e.g., days, months, years) have been clarified in all relevant sections, including the description of follow-up periods, time to diagnosis, and intervals between imaging or clinical events. We have also presented the distribution of follow-up time in Appendix Figure 1.

Reviewer 2 Report

Comments and Suggestions for Authors

The manuscript titled "Predicting PDAC Occurrence Up to 10 Years in Advance Using Features of the Main Pancreatic Duct in Pre-Diagnostic CT(MPD) Scans" addresses a potentially interesting topic. The results demonstrate that incorporating radiomic features of the MPD significantly enhances the performance of pancreatic ductal adenocarcinoma (PDAC) prediction compared to models utilizing pancreatic radiomics. The study is well-designed, and the manuscript is well-written. However, several issues need to be addressed before it can be considered for publication.

Specific Comments:

  1. Clarify the difference between diagnostic and pre-diagnostic CT scan acquisition parameters. Are there any notable differences in imaging protocols between these two cohorts? Please discuss whether the differences in acquisition protocols impacted the results.
  2. Include detailed acquisition parameters and imaging protocols for both CT and MRCP in the Methods section. This should encompass scanner models, slice thickness, reconstruction algorithms, contrast phases, and any preprocessing steps. All procedures must be described with sufficient detail to enable reproducibility by other researchers.
  3. MPD visibility on CT can be inconsistent. Please report the number of cases in which the MPD was visible versus not visible on CT images. How were non-visible MPD cases handled in the radiomic analysis? Was imputation or exclusion applied?
  4. Please provide details in discussion on how agreement or differences with MRCP were assessed.
  5. Justify the selection of the portal venous phase for CT analysis. Was this phase chosen due to optimal contrast enhancement of the pancreas or MPD? Include references supporting this choice.
  6. Discuss how anatomical variability, patient age, and tumor stage may affect MPD visibility and delineation.
  7. Figure 2: Include additional image views such as sagittal and coronal planes, in addition to axial images, to more effectively illustrate the delineation of the MPD and associated anatomical structures.
  8. Revise all figure panels and legends. Panels should be clearly labeled (e.g., (a), (b), (c), etc.), and figure legends must explicitly explain each panel and define all acronyms used.

Author Response

Thank you very much for taking the time to review this manuscript. Please find the detailed responses below and the corrections were highlighted in the resubmitted files including the manuscript and appendix files.

Reviewer 2:

The manuscript titled "Predicting PDAC Occurrence Up to 10 Years in Advance Using Features of the Main Pancreatic Duct in Pre-Diagnostic CT(MPD) Scans" addresses a potentially interesting topic. The results demonstrate that incorporating radiomic features of the MPD significantly enhances the performance of pancreatic ductal adenocarcinoma (PDAC) prediction compared to models utilizing pancreatic radiomics. The study is well-designed, and the manuscript is well-written. However, several issues need to be addressed before it can be considered for publication.

Specific Comments:

1. Clarify the difference between diagnostic and pre-diagnostic CT scan acquisition parameters. Are there any notable differences in imaging protocols between these two cohorts? Please discuss whether the differences in acquisition protocols impacted the results.

Thank you for this important point. We have reviewed the CT acquisition parameters for both the diagnostic and pre-diagnostic scans. The most CT scans in both cohorts were acquired using multiphase contrast-enhanced abdominal CT protocols. However, we acknowledge that there may be minor variations in protocol and parameters such as kVp, mAs, contrast usage, slice thickness, reconstruction algorithms. The most notably differences are slightly thinner slices and higher-resolution protocols in diagnostic scans—these did not systematically bias measurements of MPD diameter or ductal volume.

We expanded the discussion as follows: The measurements and trends in MPD diameter and volume utilizing different protocol or scanners remained consistent, suggesting that the differences in imaging protocols did not significantly affect this finding. To avoid the bias in radiomic features, we performed image harmanization and normalization before segmentation and extracting radiomic features.

2. Include detailed acquisition parameters and imaging protocols for both CT and MRCP in the Methods section. This should encompass scanner models, slice thickness, reconstruction algorithms, contrast phases, and any preprocessing steps. All procedures must be described with sufficient detail to enable reproducibility by other researchers.

Thank you for your helpful suggestion. We agree that providing detailed imaging acquisition parameters is essential for reproducibility.  We have added the related acquisition protocol and scan parameters in Table A1 including scanner models, CT parameters, MRCP parameters (e.g. field, sequence type, echo time (TE), repetition time (TR), slice thickness, and respiratory gating).

3. MPD visibility on CT can be inconsistent. Please report the number of cases in which the MPD was visible versus not visible on CT images. How were non-visible MPD cases handled in the radiomic analysis? Was imputation or exclusion applied?

Thank you for this insightful comment. We agree that MPD visibility on CT can vary, particularly in control cohort and pre-diagnostic cohort. The following is the number of cases in which the MPD was clearly visible out of the whole cohort on CT. In total, the MPD was visible in 117 out of 170 cases in control cohort, 104 out of 130 cases in pre-diagnostic cohort and 100 out of 110 cases in diagnostic cohort. The cases with non-visible MPD on CT were excluded from radiomic analysis, as accurate feature extraction requires a clearly defined MPD boundary. These data were added in line 121-124. No imputation was performed, given that radiomic features cannot be meaningfully estimated without a visible structure.

4. Please provide details in discussion on how agreement or differences with MRCP were assessed.

Thank you for your suggestion. We have now expanded the Discussion section as below. 

MRCP served as a complementary modality to CT, particularly in confirming subtle ductal changes. In cases where both CT and MRCP were performed within a close time interval (typically within 30 days), the maximum MPD diameter was compared at matched anatomical locations. Agreement was assessed using both qualitative and quantitative measures. A high correlation was observed between two modalities, with a Pearson correlation coefficient of r = 0.894, indicating strong consistency. Notable discrepancies were primarily related to ductal morphology; MRCP offered superior visualization of subtle features such as irregularities or mild dilation, which may be less apparent on CT. Nonetheless, the overall trend of increasing ductal size across disease stages was consistently observed on both imaging modalities.

5. Justify the selection of the portal venous phase for CT analysis. Was this phase chosen due to optimal contrast enhancement of the pancreas or MPD? Include references supporting this choice.

Thank you for the valuable question. We have added the rationale for selecting the portal venous phase for CT analysis in the revised Methods part and the reference. The portal venous phase was chosen because it provides optimal and consistent enhancement of the pancreatic parenchyma, which improves the contrast between the pancreatic duct and surrounding tissue, thereby facilitating accurate delineation of the MPD. This phase is also the most consistently acquired in routine abdominal imaging protocols, ensuring greater uniformity across the cohort and minimizing variability introduced by other phases.

6. Discuss how anatomical variability, patient age, and tumor stage may affect MPD visibility and delineation.

Thank you for your comments. We have expanded the Discussion to address these issues as detailed below.

Anatomical variability, patient age, and tumor stage can significantly affect MPD visibility and delineation on CT. Variations in ductal anatomy, such as branching patterns or naturally narrow ducts, may reduce contrast with surrounding tissue, complicating segmentation. MPD diameter tends to increase with age, and age-related changes like parenchymal atrophy or ductal ectasia can either improve visibility or introduce boundary ambiguity. In early or pre-diagnostic tumor stages, MPD changes are often subtle and hard to detect, while advanced tumors typically cause obstruction and upstream dilation, enhancing MPD visualization.

7. Figure 2: Include additional image views such as sagittal and coronal planes, in addition to axial images, to more effectively illustrate the delineation of the MPD and associated anatomical structures.

Thank you for this valuable suggestion. We have revised Figure 2 to include additional image views—sagital and coronal planes, alongside the axial images—to provide a more comprehensive visualization of the MPD and surrounding anatomical structures. These additional views help better illustrate MPD and improve clarity regarding its delineation. The figure legend has also been updated accordingly.

8. Revise all figure panels and legends. Panels should be clearly labeled (e.g., (a), (b), (c), etc.), and figure legends must explicitly explain each panel and define all acronyms used.

Thank you for your helpful suggestion. We have carefully revised all figure panels and legends to improve clarity and consistency. All acronyms (e.g., MPD, CT, MRCP) are now defined in the legend at first mention to ensure accessibility for all readers.